# Understanding experiences of participating in a weight loss lifestyle intervention trial: a qualitative evaluation of South Asians at high risk of diabetes

Zoe Morrison,[1] Anne Douglas,[1] Raj Bhopal,[1] Aziz Sheikh,[1,2] on behalf of the trial investigators

[1]Centre for Population Health Sciences, The University of Edinburgh, Edinburgh, UK
[2]Division of General Internal Medicine and Primary Care, Brigham and Women's Hospital/Harvard Medical School, Boston, Massachusetts, USA

**Correspondence to**
Dr Zoe Morrison;
zoemorrison@abdn.ac.uk

## ABSTRACT

**Objective:** To explore the reasons for enrolling, experiences of participating and reasons for remaining in a family-based, cluster randomised controlled trial of a dietitian-delivered lifestyle modification intervention aiming to reduce obesity in South Asians at high risk of developing diabetes.

**Design:** Qualitative study using narrative interviews of a purposive sample of trial participants following completion of the intervention. Data were thematically analysed.

**Setting:** The intervention was conducted in Scotland and resulted in a modest decrease in weight, but did not statistically reduce the incidence of diabetes.

**Participants:** We conducted 21 narrative interviews with 24 participants (20 trial participants and four family volunteers).

**Results:** Many participants were motivated to participate because of: known family history of diabetes and the desire to better understand diabetes-related risks to their own and their family's health; ways to mitigate these risks and to benefit from personalised monitoring. Home-based interventions, communication in the participant's chosen language(s) and continuity in dietitians supported their continuing engagement with the trial. Adaptations in food choices were initially accommodated by participants, although social and faith-based responsibilities were reported as important barriers to persevering with agreed dietary goals. Many participants reported that increasing their level of physical activity was difficult given their long working hours, physically demanding employment and domestic commitments; this being compounded by Scotland's challenging climate and a related reluctance to exercise in the outdoors.

**Conclusions:** Although participants had strong personal interests in participation and found the information provided by dietitians useful, they nonetheless struggled to incorporate the dietary and exercise recommendations into their daily lives. In particular, increasing levels of physical exercise was described as an additional and in some cases unachievable burden. Consideration needs to be given

## Strengths and limitations of this study

- We used narrative-based qualitative methods in a culturally appropriate way to understand the social and behavioural dimensions of a culturally adapted randomised lifestyle change trial in Scotland.
- This study highlights the need for inclusive approaches to research design.
- A limitation of this study was the use of interpreters during data generation. We sought to give participants the opportunity to participate in a language of their choosing and could not achieve this without synchronous translation.

to strengthening and supporting lifestyle interventions with community-based approaches in order to help overcome wider social and environmental factors.

## INTRODUCTION

The people of South Asian origin are up to five times more likely than White Scottish adults to develop type II diabetes and, once established, are at particularly high risk of poor outcomes.[1 2] Dietary management and physical activity have been recommended as effective contributors to the prevention of type II diabetes mellitus.[3–7] We undertook an open, family-based, cluster randomised controlled trial ('the trial') that aimed to establish the benefits and cost-effectiveness of a complex dietitian-led dietary-based and physical activity-based intervention for reducing obesity and preventing type II diabetes mellitus in people of Indian and Pakistani origin at high risk of developing diabetes living in Scotland. Participants were randomised to a

tailored intervention, comprising 15 contacts with the trial dietitians over a 3-year period, or a low-intensity intervention, comprising four visits in the corresponding period.[8]

Cultural adaption of models of care for minority populations is known to be important in promoting equal access to care and self-efficacy[9] and requires inclusion of minority groups in healthcare research.[10] Ensuring participation and retention of these populations in longitudinal studies is a concern.[11] [12] Barriers to effective engagement with minority ethnic communities include logistical considerations,[13] language-related barriers[14] and issues of trust and respect between researchers and marginalised communities.[14] [15] Extensive cultural adaptations were built into the trial intervention,[16] specifically flexible home-based interventions by the same dietitian, multilingual written resources and verbal communication, and modification of high-calorie traditional dishes.

The majority of recruitment to the trial was achieved informally through personal communications based on trusted relationships such as faith-based activities and social groups including lunch clubs, referral by friends and general practitioners, community locations such as specialist food and clothing shops.[12] Participant retention rate within the trial of 97.7% far exceeded expectation. Outcomes of the trial were variable for individual participants. Overall the tailored intervention led to weight loss ≥2.5 kg for 33 participants compared with 12 in the control group. The intervention group spent 3 h per week on physical activity compared with the control group's 2.1 h, with little difference in food shopping and food preparation time. These results are reported in full separately.[17] This paper reports findings from a qualitative study of participant experiences of the trial. As the main trial was a lifestyle intervention utilising quantitative parameters we sought to investigate the social and behavioural aspects of the trial that were not addressed in the main trial outcome measures. We aimed to obtain a rich and multifaceted understanding of reasons for trial participation and retention, and factors influencing adherence to the lifestyle intervention to give insight into the social and behavioural dimensions of the trial method and results.

## METHODS

We undertook a qualitative study[18] to inform our understanding of patient experiences of the trial processes.[19] We used narrative-based interviewing,[20] a method that encourages participants to tell their story as they choose, in preference to interviews that utilise structured topic guides. Asking participants to tell their story allows people to spontaneously construct an account of their own experiences in a manner that is strongly shaped by and reflective of their cultural preferences. This method has been found to be an effective method of generating and interpreting experience-centered, culturally orientated data with people of South Asian backgrounds.[21]

The methods for the main trial, including participant inclusion and exclusion criteria, the nature of the randomisation procedure and retention rate are all described in detail separately.[17] A summary CONSORT flow chart is shown at figure 1 as contextual information for this study. Details of the trial participant population are shown in table 1. We focused on trial participants and family volunteers exiting the study. We utilised purposeful sampling[22] to ensure representation of the diversity within the trial population by sex, ethnicity, faith group, geographical location (Glasgow and Edinburgh) and whether they were allocated to the intervention or control group.[8]

During their final trial visit, participants were invited by their dietitian to take part in a further piece of related research and given an information sheet regarding the qualitative study. The trial dietitians then obtained consent to be contacted from those who were willing to consider participation. A selection of participants was made utilising the purposeful sampling strategy and contacted to arrange a research meeting, at the beginning of which the qualitative study was explained again before written consent to take part was obtained by the researcher. All interviews were conducted at a location and in the language of the participants' choice; interpreting services were used, if necessary. Interviews lasted between 1 and 2 h and in all but one case took place in the participant's home. Each of the family volunteers who took part was related to one of the 20 trial participants and was interviewed together with their family member. All fieldwork was undertaken with due regard to maintaining the best interests of participants and, in particular, assuring their confidentiality and anonymity. The researcher was blinded to the performance of trial participants throughout the course of the research process, including study design, data generation and analysis.

Narrative methods seek to gather participant accounts of their experiences in their own words in the form of stories by using natural prompts to encourage communication rather than questions to stimulate response.[20] We used preliminary qualitative work (unpublished) to identify open questions to prompt participants' stories, when necessary, including for example:

▶ What were the motivations, perceptions and attitudes that led you to express an interest and then agree to participate in this trial?
▶ What factors have influenced your ability to engage with, and be faithful to, the study interventions?
▶ What factors have influenced participants' decisions to remain involved in the trial?
▶ More generally, what were the most memorable events and experiences that influenced decisions to accept the invitation for screening, agree to participate, adhere to and remain enrolled in the trial?

 Morrison Z, Douglas A, Bhopal R, *et al. BMJ Open* 2014;**4**:e004736. doi:10.1136/bmjopen-2013-004736

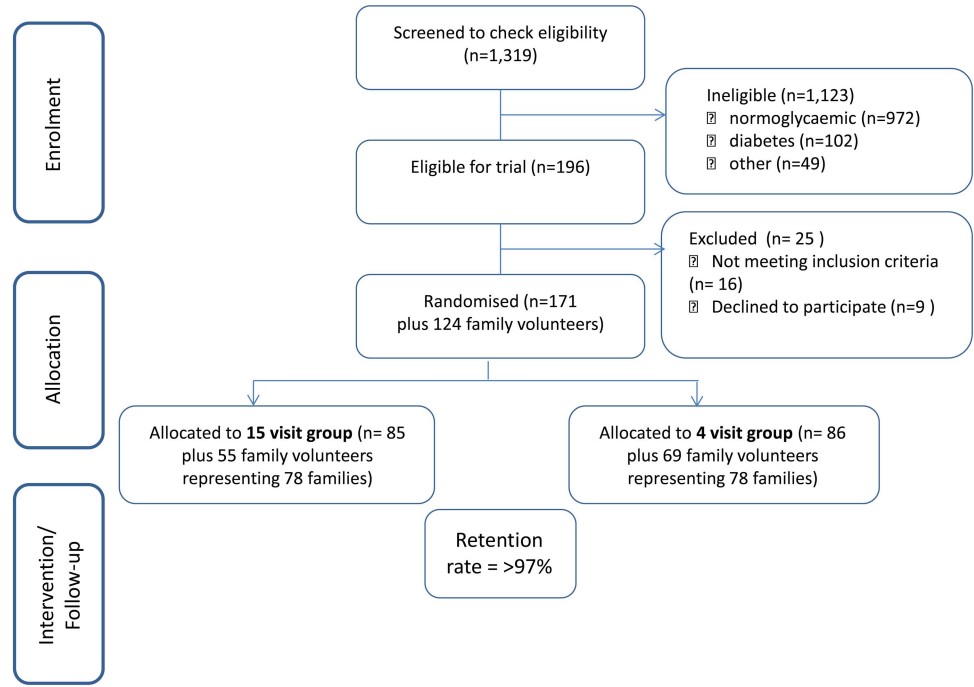

**Figure 1** Main trial summary consort flow chart.

Thematic data analysis[20] was concurrently undertaken using NVivio9 software,[23] allowing emerging themes to inform on-going data collection using the constant comparison method.[24] We actively considered alternative explanation cases and allowed for the researcher's reflexive analysis of interpretations.[20] Regular discussions of the emerging findings among the trial qualitative subgroup and the active seeking of disconfirming data further ensured the trustworthiness of findings. Data collection ceased when saturation could reasonably be assumed.[25] Fully anonymised study results were only shared with trial dietitians once the trial was closed.

## RESULTS

We conducted 21 narrative interviews with 24 participants (20 trial participants and four family volunteers). We achieved high quality data from a diverse sample, enabling spontaneous account of participants'

experiences in their chosen language(s) and narrative structure. Our process of recruitment and data generation is shown in figure 2. Qualitative study participants are summarised by geographical location in table 2, by intervention, ethnicity, language and faith in table 3 and detailed in full in table 4. Findings are reported in conjunction with illustrative data in relation to experiences of participation and retention within the trial, and adherence to the intervention. Our analytical framework is summarised in box 1. We did not identify significant points of differentiation between the intervention and control group participants. We did identify a small number of gender-specific considerations.

### Participation and retention in the trial

Perceived benefits of participation in the trial, the accommodation of participant choice of language and location, and trusting relationships between trial investigators and participants contributed to the trial's very high retention rate.

#### Perspectives on potential benefits of participation

Participants could not always fully recall how they were recruited into the trial. Two participants described themselves as having joined the trial because they were *on the borderline* of having diabetes (I.15,18) and three specifically noted their desire to lose weight (I.8,13,19).Some participants had been attracted by the provision of health-related information (I.3,5,6,7,10,12). Increased awareness was particularly attractive in the light of family histories of diabetes mellitus (I.4,10,13,14,15,17,19,11).

| Table 1 | Participant population by geographical location | | | | | |
|---|---|---|---|---|---|---|
| | **Glasgow** | | **Edinburgh** | | **Total** | |
| Research participants | 132 | | 39 | | 171 | |
| | Male | Female | Male | Female | Male | Female |
| | 58 | 74 | 20 | 19 | 78 | 93 |
| Family volunteers | 114 | | 10 | | 124 | |
| | Male | Female | Male | Female | Male | Female |
| | 28 | 86 | 0 | 10 | 28 | 96 |

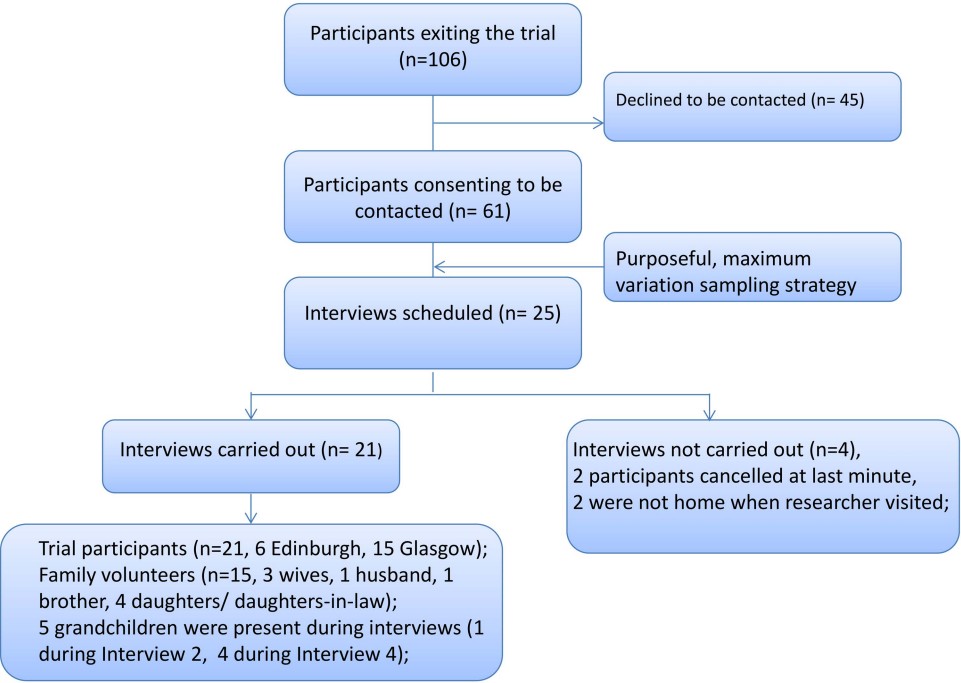

**Figure 2** Qualitative study recruitment and data generation.

My parents had diabetes and my father-in-law also had, so I have always had fear about it. In fact I consulted libraries also to gain knowledge about it. (I.11)

Other participants were in contrast more ambivalent about the trial, participating on the basis of *no harm* rather than in pursuit of direct benefit (I.1,6,9).

## Participant choice

The emphasis on patient choice built into the trial design contributed to participant engagement and retention. Accommodating preferences for consultation at home or in clinic was appreciated as some welcomed the opportunity to get out (I.10, 15) while others welcomed home consultations (I.4,9,17). However, not every participant kept a diary and home visits presented some scheduling difficulties that required out-of-hours working by dietitians, as one participant observed:

You can't do it all the time because the people are at home at different times of the day, some are in the evening, some late evening, they are not there during the day time. (I.9)

Being able to speak to female dietitians in their language(s) of choice was valued by female participants (I.4,7,11,13), with participants feeling free to take the approach best suited to them:

It made me—like—relaxed when she was around because I didn't, sometimes I don't understand in English, then I could speak to her in Indian. But mostly I didn't want to speak to her in Indian. (I.16)

Such preferences were less frequently expressed by men (I.18,19).

## Relationships

Some participants decided to participate in the trial as a result of knowing members of the research team. In addition, the relationships that developed between participants and their dietitians over the course of the trial positively contributed to retention. While participants did not commonly refer to or produce for discussion their written materials, they placed importance on good verbal communication (I.11,16), empathy and compassion (I.7,10,18).

She used to advise me but so compassionately and afterwards I used to think over it that it is for my help ultimately I will be the one who will benefit she is not my relative still she cares so much. (I.19)

**Table 2** Qualitative study participants by geographical location

| | Glasgow | | Edinburgh | | Total | |
|---|---|---|---|---|---|---|
| Research participants | 14 | | 6 | | 20 | |
| | Male | Female | Male | Female | Male | Female |
| | 9 | 6 | 4 | 2 | 13 | 7 |
| Family volunteers | 4 | | 0 | | 4 | |
| | Male | Female | Male | Female | Male | Female |
| | 1 | 3 | 0 | 0 | 1 | 3 |

**Table 3** Qualitative study participants by intervention, ethnicity, language and faith

| | Trial participants | Randomised group | | Ethnicity* | | Preferred language* | | | | Use of interpreter during interview | Faith* | | | Family volunteers | |
|---|---|---|---|---|---|---|---|---|---|---|---|---|---|---|---|
| | | 15 intervention | 4 intervention | Indian | Pakistani | English | Urdu | Punjabi | More than one language | | None | Muslim | Sikh | 15 intervention | 4 intervention |
| Male | 12 | 7 | 5 | 6 | 6 | 9 | 0 | 0 | 3 | 3 | 1 | 6 | 5 | 0 | 1 |
| Female | 8 | 2 | 6 | 3 | 5 | 3 | 2 | 1 | 2 | 4 | | 4 | 4 | 1 | 2 |
| Total | 20 | 9 | 11 | 9 | 11 | 12 | 2 | 1 | 5 | 7 | 1 | 10 | 9 | 1 | 3 |

*Self-defined.

## Adherence to the intervention

Adherence to the trial proved a complex issue to investigate as participants were well informed and reported the correct behaviours even if they were not pursuing these. Adherence involved different understandings of the advocated lifestyle changes and unanticipated motivations such as caring for other family members or a sense of obligation to the dietitian. Non-adherence to the trial was in some case culturally motivated, or due to contextual factors such as climate or lack of time.

### Understanding adherence

Participants who felt they had adhered to the trial mostly described changes they had made in relation to frequency of exercise (I.11,12) and more informed food choices (I.4,10,11,17,18,21). Some felt that this had not been a particularly significant change as they had a balanced diet already (I.9,12). Conversely, some were evangelical about the trial and their related achievements. Some had lost significant amounts of weight (eg, *half a stone* (I.11)). Two men described how their lives had fundamentally changed, one having given up high alcohol consumption (I.15), the other having become a runner for charity:

> I could have never imagined …I could not even walk 2 kilometres then I have started doing 5 [km] then 10 [km] then I have done lot of 10 then I thought lets go for the bigger [half marathon]then it took me 2 hours and 19 minutes …So with that time we raised £2,000—I was in the paper! (I.19)

Many participants had difficulty in sustaining lifestyle changes (I.10,16,13,14) describing a process of *fits and starts* (I.10) as they struggled consistently to follow the advice given despite known health risks. This was true even subsequent to a diagnosis of diabetes (I.7,16). In contrast, looking after other family members influenced adherence. One participant attributed a weight loss of 8 kg in part to different methods of food preparation (eg, less frying) and their having increased their daily exercise due to the need to look after their son's dog (I.8). Another described how they had followed the intervention because they felt it would be beneficial to the whole family (I.11).

Scheduled consultations themselves impacted on levels of adherence as participants either avoided their appointments as they were on the *downward slope* (I.10,11) or adapted their behaviour in anticipation:

> She used to come in every three months so before she used to come I used to feel a kind of pressure to make sure that my weight had not increased. (I.18)

### Reasons for non-adherence

Culture considerations were evident in discussions regarding the challenges of adherence to the lifestyle intervention. The importance of food within South Asian cultures was referred to for example in the

**Table 4** Details of interview participants

| Interview no. | Location | Randomised group (1=tailored intervention, 2=low intensity intervention) | Gender (M/F) | Ethnicity I=Indian P=Pakistani | Preferred language for speaking | Number of participants (family volunteer) | Researcher |
|---|---|---|---|---|---|---|---|
| 1 | Glasgow | 2 | M | I | Punjabi | 1 | ZM (and interpreter) |
| 2 | Glasgow | 2 | F | I | Punjabi | 1 | NC (and interpreter) |
| 3 | Edinburgh | 1 | M | P | English | 1 | NC |
| 4 | Glasgow | 2 | F | P | Urdu | 1 | NC |
| 5 | Edinburgh | 1 | F | P | English | 1 | NC |
| 6 | Glasgow | 2 | F | I | Punjabi | 1 | NC (and interpreter) |
| 7 | Glasgow | 2 | F | I | Punjabi | (Wife) | NC (and interpreter) |
| 8 | Glasgow | 1 | M | I | English | 1 | NC |
| 9 | Edinburgh | 2 | M | P | English | 1 | NC |
| 10 | Glasgow | 1 | M | I | English | 1 | NC |
| 11 | Glasgow | 2 | F | P | Urdu, Hindi, Punjabi | 1 | NC |
| 12 | Glasgow | 2 | M | I | English | 1 | NC |
| 13 | Glasgow | 2 | F | P | English | 1 | NC |
| 14 | Edinburgh | 1 | M | P | English | 1 | ZM |
| 15 | Edinburgh | 1 | M | I | English | 1 | ZM |
| 16 | Edinburgh | 1 | F | I | English | 1 | ZM |
| 17 | Glasgow | 1 | M | P | English | 1 | ZM |
| 18 | Glasgow | 1 | M | P | Urdu, Punjabi | 2 (Wife) | ZM (and interpreter) |
| 19 | Glasgow | 2 | M | I | English | 2 (Brother) | ZM |
| 20 | Glasgow | 2 | M | P | Urdu | 1 | ZM (and interpreter) |
| 21 | Glasgow | 2 | M | P | English, Urdu | 2 (Wife) | ZM (and interpreter) |

context of sharing home-made sweets with grandchildren (I.2,14) and traditional food preparation techniques:

Once a week they have children all come so we feel that the food should be much nicer according to the tradition and also children don't like ordinary vegetables they fancy food like from McDonald's so just to compete with that kind of food we try to make our old Indo-Pakistani dishes. (I.20)

The role of food in community functions (I.15) and faith were also given as challenges to adherence, with

**Box 1   Summary of themes within the data**

Participation and retention
▶ Perspectives on potential benefits of the participation;
▶ Participant choice;
▶ Relationships.
  Adherence to the intervention
▶ Understanding adherence;
▶ Reasons for non-adherence.

one participant feeling she could not avoid gaining weight:

I know that if I lose more weight I'm going to put it back on at Ramadan because I just have to walk by food and it just adds onto my body, the fat just is invisible and it runs after you. (I.13)

While the trial was designed to be culturally sensitive to the needs of South Asian populations, some members of the population felt that the advice did not accommodate their more international food preferences (I.10,13,15).

Other reasons for non-adherence related to wider contextual factors. Participants felt they were too busy to find the time to exercise (I.1,4,14,15,16). Two male participants noted the physical demands of their employment as a barrier to exercise (I.14,17), while some female participants felt unable to get enough time out of the house (I.2):

I try to go for walks but it's hard to find time as I have so many grand-children and women in our community

don't get out of the house that much. There is no one else who will come with me. (I.4)

The climate was a consideration for South Asians living in Scotland as participants found recommended exercise such as walking problematic in the cold (I.1,2,15,16) and food was described as a cultural representation of warmth (I.2). It therefore seemed counterintuitive to increase one without the other.

## DISCUSSION

We used qualitative methods to understand the social and behavioural dimensions of the first family-based, culturally adapted randomised lifestyle change trial to investigate approaches to reducing obesity and prevent type II diabetes mellitus in people of Indian and Pakistani origin living in Scotland. Findings indicate that individuals may not necessarily prioritise their own health over other factors when making lifestyle choices despite known risks and prevention strategies. Participants were attracted to the trial by the availability of regular health monitoring and personalised information, particularly those who were aware of a family history of diabetes. The choice of home-based or clinic-based interventions, communication in the participants' chosen language(s) and trusting relationships between participants and dietitians contributed to the high retention rate. Participants stated that they did not find adherence to advice on food choices and preparation problematic, although some conceded that they did not always choose to take the advice, citing examples of community and faith-based considerations that made consistent adherence difficult. Participants did find increasing their levels of physical activity problematic given demands such as the Scottish climate, long working hours, physically demanding jobs and domestic commitments.

One important limitation of this study was the use of interpreters during data generation. We sought to give participants the opportunity to participate in a language of their choosing and could not achieve this without synchronous translation. During the initial stages of the study retrospective quality checking of interview audio and transcripts highlighted that the interpreter was not accurately reporting the views of the participants. We addressed this by recruiting a new interpreter and providing training as to the exact nature and purpose of the services required. To ensure that the data from the three affected interviews were not lost we had the original audio recording retranslated. We note, however, that the conduct of these interviews was compromised to a small extent by this limitation. A further limitation is the lack of inclusion of Indian Hindu participants and the small sample of family volunteers. For practical reasons (such as interview cancellation by participants) we could not achieve this within our sample during the time available.

In discussing our findings in the light of current literature on participation and retention we consider the benefits

sought from the intervention, flexibility within the trial design and participant/dietitian relationships. Regarding adherence our participants talked about lifestyle changes and we have reported findings accordingly. In contrast, the literature considers attitudes to food and physical activity separately and we discuss our findings in this context.

The need for individualised and context-specific information for people with prediabetes has been acknowledged[26] and was a benefit participants sought from engaging with the trial. People recognised they were at risk due to family histories. Losing weight was not articulated as a dominant sought-after benefit even though it was a secondary outcome of the trial and this may be due to culturally situated traditional beliefs about body size. In South Asian communities a tendency to equate large size with healthy womanhood, marriage and reproduction, securing a good job and high social status has been identified,[27] although this view may be less prevalent now. Understanding of the causal relationship between weight and diabetes was less evident, reiterating calls for education within South Asian communities on the causes and prevention of diabetes.[28]

Findings highlight the importance of accommodating participant choice in the design of complex lifestyle interventions. Appreciation of flexibility within diabetes prevention programmes has been recognised,[29] as has the need to overcome known barriers to effectively engaging with minority ethnic communities.[13–15 30] While culturally sensitive adaptations are essential and were achieved within this trial[16] they should avoid reinforcing stereotypes, for example by focusing on traditional dishes. This did not necessarily sufficiently accommodate complex patterns of dietary acculturation.[31]

Despite cultural adaptation, the trial written resources were rarely referred to. In contrast, verbal communication and participant/dietitian relationships were repeatedly referred to. The importance of non-judgemental professional attitudes, and empathy within self-management interventions has been recognised[26 29] and, together with known issues of trust and respect between researchers and marginalised communities,[14 15] was anticipated in the trial design. The intimate nature of this home-based lifestyle intervention required a compassionate approach to investigation, out-of-hours working and an emphasis on participant choice different to that of the majority of interventions delivered in clinical settings. Dietitian skill in building relationships was felt to be very important in addition to skills in the verbal communication of advice and information. The importance of continuity in dietician staff was emphasised in trial design, but the strength of participant–dietitian relationships was not anticipated and proved to be significant in retention across the trial (99% in the intervention group and 97% in the control group). In some cases it seemed that participant loyalty was to the dietitian rather than the trial. This indicates the importance of relational aspects of trial participation that would benefit from further investigation.

Our findings suggest that the cultural construction of food made it difficult for South Asians to make the healthy food choices advised despite known health risks. For members of this community, food is the focus of good living and etiquette.[27] In communities that have previously faced economic insecurity, being able to afford food is a sign of success, demonstrating social power and hospitality.[27 31 32] Although most migrants alter their eating habits following migration,[31] South Asian systemic beliefs regarding the heating and cooling properties and merits of food endure.[33] The role of food in community-based and faith-based activities together with complex intergenerational patterns of dietary acculturation detracted from self-reported consistent adherence to the tailored and low-intensity interventions.

It is known that the South Asian population has lower levels of physical activity than the general population and lesser intrinsic motivations for physical exercise.[34 35] We have corroborated proposed reasons for this, including lack of time and/or childcare[34 35] and gendered barriers to exercise, such as the availability of single-sex facilities and the depiction of exercise as a selfish act detracting from family care.[35] We further note the role of climate and the interplay between South Asian systemic beliefs regarding the heating and cooling properties and merits of food[33] and the demands of living and undertaking physical activity in the cold Scottish climate.

Our study considered one research subpopulation: the South Asian community living in Scotland. A more general insight from these findings is that well-informed individuals may not necessarily choose or feel able to act on health-related advice due to other lifestyle considerations. People sought health-related information as the main benefit of participation and the dietary requirements of the trial were not felt by participants to be an extraburden in their everyday life. However, culturally situated influences on food choices often took precedence over the advice and information given in intervention and control groups despite acknowledged health implications. This highlights the need for the adaptation of lifestyle interventions to accommodate cultural considerations among ethnic minority research participants.

The increased levels of physical activity were, however, seen to pose an additional burden by participants, again due to cultural factors including those previously identified, such as lower intrinsic motivation, a lack of time and/or access to appropriate facilities and domestic considerations, each and all of which may impact on the prevention and management of diabetes. In addition, findings highlighted the role of climate in decisions relating to food and physical activity. This suggests that diet and exercise are less distinct within South Asian communities and indicates a need to better understand the impact of climate on lifestyle choices for immigrant populations.

**Acknowledgements** The authors have previously thanked and named many individuals who contributed to the trial. (8) The authors further acknowledge the Trial Dietitians (Alyson Hutchison, Anu Sharma, Sunita Wallia and Ruby Bhopal), Nandini Choudhuri, Sujama Roy and Salman Aziz; current Trial Steering Committee (Professor Nigel Unwin (ex-chair), Prof Graham Hitman (chair), Dr Nita Forouhi, Dr Deepak Bhatnagar, Dr Ghada Zoubiane and Mr Iqbal Anwar); the trial Data Monitoring and Ethics Committee (Professor Iain Crombie, Dr Mike Small, Dr Mike Kelly, Professor Kamlesh Khunti); South Asian Community and religious organisations and individuals who contributed to the recruitment efforts; and finally to all the participants who gave their time to take part in this study.

**Collaborators** Trial investigators: John F Forbes, Jason M R Gill, Michael Lean, John McKnight, Gordon Murray, Naveed Sattar, JaakkoTuomilehto and Sarah H Wild.

**Contributors** ZM and AS were responsible for the study design, supervised the data collection and drafted the manuscript. AD and RB designed the protocol for the larger study and assisted with obtaining consent for data collection together with other members of the trial team. Each of the authors was involved in data analysis and interpretation, critical review and refinement of the manuscript.

**Funding** This study was funded by the National Prevention Research Initiative (G0501310), a funding consortium comprising the British Heart Foundation; Cancer Research UK; Department of Health; Diabetes UK; Economic and Social Research Council; Medical Research Council; Health & Social Care Research & Development Office for Northern Ireland; Chief Scientist Office, Scottish Government Health Directorate; the Welsh Assembly Government; and World Cancer Research Fund. Additional financial support was provided from NHS Lothian and NHS Greater Glasgow & Clyde R&D, Chief Scientist Office, NHS Health Scotland and NHS National Services Scotland.

**Competing interests** None.

**Ethics approval** MREC.

**Provenance and peer review** Not commissioned; externally peer reviewed.

**Data sharing statement** Interview audio recordings and transcripts are available from the corresponding author at the University of Edinburgh, who will provide a permanent, citable and open access home for the dataset.

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
