## [Reviewer comments · BMJ Open]

Some articles will have been accepted based in part or entirely on reviews undertaken for other BMJ Group journals. These will be reproduced where possible.

ARTICLE DETAILS

TITLE (PROVISIONAL)	Understanding experiences of participating in a weight loss lifestyle intervention trial: qualitative evaluation of South Asians at high risk of diabetes
AUTHORS	Morrison, Zoe; Douglas, Anne; Bhopal, Raj; Sheikh, Aziz

VERSION 1 – REVIEW

REVIEWER	Ambady Ramachandran India Diabetes Research Foundation and Dr. A. Ramachandran's Diabetes Hospitals, Chennai, India
REVIEW RETURNED	01-Mar-2014

GENERAL COMMENTS	This is a qualitative study using narrative interview with samples of trial participants on a lifestyle modification intervention in South Asians aiming to reduce obesity. The paper does not clearly define the inclusion/exclusion criteria, the randomisation procedure, the criteria for control and study group. What is meant by 'completing the trial'? In fact, the calculations show a retention rate of >97%. The reviewer is unable to understand how it is so high, when only 21 out of 61 participants took part in the interview. * How were the 21 cases selected for the interview ?* How could the results of interviews of a selected few be generalised to the total study population.* In general, the presentation of the methodology is not adequate to make the reader understand the study protocol and the conclusions drawn from the study.
--

REVIEWER	Harsimran Singh University of Virginia, USA
REVIEW RETURNED	06-Mar-2014

GENERAL COMMENTS	The topic discussed in the manuscript is of huge importance given that we do not very much about factors affecting participation of minority populations in clinical trials. The reported study includes experiences of participants who had just completed a study, which is far more authentic compared to merely asking people (who have not participated in a trial) what factors might affect their motivations to participate and remain in a trial. Below are a few comments for the authors to consider in terms of editing and improving the manuscript: (1) In the article summary - for strengths, it has been stated that family-based qualitative methods were used. However, only 4 family
---

	volunteers were included in the study while there were 20 trial participants. I am aware that qualitative methodology should not emphasize on numbers as much as the content. But with only four family volunteers, I think it is inaccurate to state it as the strength of the study. Perhaps if you had about 10 family members, it would be a different story. Text should be revised to include this point. (2) Methods - It would be helpful to have a couple of examples of questions that were asked of the participants - for the reviewers to better understand the process of narrative-based interviewing (in relation to structured interviews). (3) It was not clear to me where the interviews were held, how long did they last? Regarding the four family volunteers - were they related to any of the 20 trial participants? If so, were the participants present when the volunteers were interviewed? This and other related information is important to better understand the context in which these interviews were held. (4) While selecting participants for these interviews, was the researcher blinded to the performance of trial participants? If not - then it would be important to comment on how this may have affected the content of the interview (more successful vs less successful participants). (5) The authors mention the strength of the participant - dietician relationships as an important factor. It would be interesting to know if the dietician(s) were of participants' ethnicity? If so, it should be discussed.
--	--

REVIEWER	Dr Cathy E. Lloyd The Open University, UK
REVIEW RETURNED	18-Mar-2014

GENERAL COMMENTS	Thank you for asking me to review this interesting paper. Overall it is well written, although there are a few typos and one or two sentences/phrases repeated. Examples include: Page 5 - under Relationships – the third 3rd sentence needs clarifying. Page 8, line 21 - take out 'upon' Page 8, line 52 - repeats line 41 There are a number of clarifications that are required as follows: The authors state that interpreting services were used when required which is perfectly acceptable. Given the problem with the first interpreter - was the data that was collected by this interpreter discarded? If not then how was the data clarified or the quality improved and what limitations does this bring to the study? On page 5 one participant states that she did not understand English but didn't want to speak Indian - can the authors clarify how this interview proceeded given this participant's concerns? Preference for language less likely to be expressed by men - why was this? Can the authors clarify exactly how the research questions for this study were generated? This paper suggests that the participants felt that the dietary
--

	requirements of the intervention were not perceived to be a burden but unless I'm mistaken it appears they were not acted upon. In contrast exercise was perceived to be a burden. Furthermore, the authors suggest that a range of culturally specific factors could explain these findings, including lack of time, low intrinsic motivation and so on. I wonder how these perceived barriers and notions of burden differ from or are culturally specific to Pakistani and Indian individuals rather than general barriers for anyone taking part in a trial such as this? This paper does have implications for practice, but also implications for research. I would ask the authors to unpick both of these more and suggest potential ways forward for carrying out this type of research as well as making recommendations for practice. Along the same lines, there appears to be some implications in terms of the need for a flexible intervention style. For example, the very high retention rate appears to be mostly due to the strong relationship between the participant and the dietician. What implications does this have for 1) research and 2) practice? By addressing these questions I think the authors can strengthen their case for their work making a contribution to the field (which I am sure it does – it's just not explicit at the moment).
--	--

VERSION 1 – AUTHOR RESPONSE

Reviewer: 1

The paper does not clearly define the inclusion/exclusion criteria, the randomisation procedure, the criteria for control and study group. What is meant by 'completing the trial'? In fact, the calculations show a retention rate of >97%. The reviewer is unable to understand how it is so high, when only 21 out of 61 participants took part in the interview.

Response: We apologise that this confusion arose because of our failure to adequately distinguish the qualitative work from the findings of the main trial. We have now amended the Introduction to make explicit the difference between the main trial and the qualitative study reported in this manuscript. The Methods section of the paper has now been revised so that it focuses exclusively on our qualitative study. The revised Methods section now makes clearer the inclusion and exclusion criteria for entry into the qualitative study and the proportion of those that agreed to participate. Given that the trial results are already published,¹ we have now more explicitly referred interested readers to this report where issues relating to trial inclusion and exclusion criteria, the nature of the randomisation procedure and retention rate are described in detail.

How were the 21 cases selected for the interview?

Response: This is described in the first paragraph on p.5, where we describe our purposeful sampling strategy and the variables used to construct our sampling matrix (shown in Table 1). We now also note that purposeful sampling informed our participant selection (para.2, p.5).

How could the results of interviews of a selected few be generalised to the total study population?

Response: Our strategy of enquiry derives from the qualitative paradigm in social science and is designed to allow an in-depth understanding of complex social phenomenon from differing perspectives and alternate viewpoints. As is the case with all qualitative research, our findings are not offered as generalizable facts, but rather as nuanced descriptions that allow the generation of transferable insights.²

Reviewer: 2

In the article summary - for strengths, it has been stated that family-based qualitative methods were used. However, only 4 family volunteers were included in the study while there were 20 trial participants. I am aware that qualitative methodology should not emphasize on numbers as much as the content. But with only four family volunteers, I think it is inaccurate to state it as the strength of the study. Perhaps if you had about 10 family members, it would be a different story. Text should be revised to include this point.

Response: We concur with this point and have removed mention of this as a strength from the revised manuscript.

Methods - It would be helpful to have a couple of examples of questions that were asked of the participants - for the reviewers to better understand the process of narrative-based interviewing (in relation to structured interviews).

Response: We have added the following sentence to the methods section (para.2, p.5) to aid understanding of the interview process:

“Narrative methods seek to gather participant accounts of their experiences in their own words in the form of stories by using natural prompts to encourage communication rather than questions to stimulate response.”

We then go on to give some examples of the prompts utilised.

It was not clear to me where the interviews were held, how long did they last? Regarding the four family volunteers - were they related to any of the 20 trial participants? If so, were the participants present when the volunteers were interviewed? This and other related information is important to better understand the context in which these interviews were held.

Response: We have added the following description into the revised Methods section (para.1, p.5): “All interviews were conducted at a location and in the language of the participants’ choice; interpreting services were used, if necessary. Interviews lasted between one and two hours and in all but one case took place in the participant’s home. Each of the family volunteers who took part was related to one of the 20 trial participants and was interviewed together with their family member.”

While selecting participants for these interviews, was the researcher blinded to the performance of trial participants? If not - then it would be important to comment on how this may have affected the content of the interview (more successful vs less successful participants).

Response: We have augmented the method section for clarification as follows (para.1, p.5): “The researcher was blinded to all primary and secondary outcome data for the randomised trial throughout the course of the research process, including study design, data generation and analysis.”

The authors mention the strength of the participant - dietician relationships as an important factor. It would be interesting to know if the dietician(s) were of participants' ethnicity? If so, it should be discussed.

Response: The trial employed both South Asian and White dieticians and they were therefore in some cases matched, but not in others. Ethnicity was not referred to by participants as an influence in the conduct of the study. Rather, the more important contributing factors to the participant-dietician relationships were the ability to accommodate different languages and participant choices regarding the scheduling of interventions, compassionate and empathetic relationships and staff continuity, as reported (para.3, p.8).

Reviewer: 3

There are a few typos and one or two sentences/phrases repeated. Examples include:

Page 5 - under Relationships – the third 3rd sentence needs clarifying.

Page 8, line 21 - take out 'upon'

Page 8, line 52 - repeats line 41

Response: We apologise for these typos, which have now been corrected.

The authors state that interpreting services were used when required which is perfectly acceptable. Given the problem with the first interpreter - was the data that was collected by this interpreter discarded? If not then how was the data clarified or the quality improved and what limitations does this bring to the study?

Response: We have now clarified this point (para. 6, p.8) with the following explanation:

“Although there were difficulties, these interviews still yielded important data. To maximise the usefulness of the data from the three affected interviews, we had the original audio recordings re-translated. We note however that these interviews may to an extent have been compromised by the problems with interpretation.”

On page 5 one participant states that she did not understand English but didn't want to speak Indian - can the authors clarify how this interview proceeded given this participant's concerns?

Response: This interview was conducted with a bi-lingual participant in English at her request. A strength of this study was the ability to accommodate participant choice, in some cases involving conducting the interview in more than one language (i.e. a combination of English and another language).

Preference for language less likely to be expressed by men - why was this?

Response:

The relative lack of emphasis in relation to choice of language by men is we believe likely to have been because they tend to be more employed in work outside the home, which had given them opportunities to develop their English language skills. Please note that this is speculative on our part as we did not formally investigate this as a research question in our study.

Can the authors clarify exactly how the research questions for this study were generated?

Response: The research questions were generated by a multi-disciplinary group of investigators through drawing on the existing literature and identifying key questions in relation to participants' perspectives and experiences of trial processes that would not have been answered through the more quantitative parameters being investigated through the formal trial analysis. We have added to the final paragraph of the introduction to better describe this work (para. 3, p.4):

“As the main trial was a lifestyle intervention utilising quantitative parameters we sought to investigate the social and behavioural aspects of the trial that were not addressed in the main trial outcome measures.”

This paper suggests that the participants felt that the dietary requirements of the intervention were not perceived to be a burden but unless I'm mistaken it appears they were not acted upon. In contrast exercise was perceived to be a burden.

Response: The researcher was blinded to the results of the main trial and it is not known whether the dietary requirements were acted upon by participants. In general the dietary advice received resonated with participants and they described how they were able to operationalise this into their lives to varying extents.

Furthermore, the authors suggest that a range of culturally specific factors could explain these findings, including lack of time, low intrinsic motivation and so on. I wonder how these perceived barriers and notions of burden differ from or are culturally specific to Pakistani and Indian individuals rather than general barriers for anyone taking part in a trial such as this?

Response: We discuss our findings in relation to the challenges faced by members of the general

population and go on to discuss some more specific issues facing ethnic minority groups. We have amended the text to make this distinction more explicit (penultimate paragraph, p.10).

This paper does have implications for practice, but also implications for research.

I would ask the authors to unpick both of these more and suggest potential ways forward for carrying out this type of research as well as making recommendations for practice. Along the same lines, there appears to be some implications in terms of the need for a flexible intervention style. For example, the very high retention rate appears to be mostly due to the strong relationship between the participant and the dietician. What implications does this have for 1) research and 2) practice? By addressing these questions I think the authors can strengthen their case for their work making a contribution to the field (which I am sure it does – it's just not explicit at the moment).

Response: We appreciate this positive suggestion and have made some small additions to the discussion section of our paper to make our contribution to the field more explicit. This paper has implications for the design of lifestyle interventions including cultural awareness, the importance of the continuity of relationships between researchers and participants and a flexible approach, emphasising communication and the accommodation of different linguistic preferences. It also points to important practical considerations, including the need to avoid stereotypes when providing advice on diet and exercise, and the need to appreciate cultural factors within lifestyle interventions.

The opportunity to respond to the reviewers' constructive suggestions has helped us to improve our work and for this we are very grateful. We trust that these revisions are to your satisfaction. Please do not however hesitate to contact me should you require any further revisions or clarification.

VERSION 2 – REVIEW

REVIEWER	Harsimran Singh University of Virginia Health System, USA
REVIEW RETURNED	20-May-2014

GENERAL COMMENTS	The authors have done a thorough job of understanding my concerns and I would be happy to accept this manuscript for publication.
---